# Sensing Leakage of Electrolytes from Magnesium Batteries Enabled by Natural AIEgens

**DOI:** 10.3390/ijms231810440

**Published:** 2022-09-09

**Authors:** Yingxiang Zhai, Jiguo Zhang, Jian Li, Shouxin Liu, Zhijun Chen, Shujun Li

**Affiliations:** Key Laboratory of Biobased Material Science and Technology of Ministry of Education, Northeast Forestry University, Hexing Road 26, Harbin 150040, China

**Keywords:** laccaic acid, aggregation-induced emission, smart film, detect of Mg^2+^

## Abstract

The potential for leakage of liquid electrolytes from magnesium (Mg) batteries represents a large hurdle to future application. Despite this, there are no efficient sensing technologies to detect the leakage of liquid electrolytes. Here, we developed a sensor using laccaic acid (L-AIEgen), a naturally occurring aggregation-induced emission luminogen (AIEgens) isolated from the beetle Laccifer lacca. L-AIEgen showed good selectivity and sensitivity for Mg^2+^, a universal component of electrolytes in Mg batteries. Using L-AIEgen, we then produced a smart film (L-AIE-F) that was able to sense leakage of electrolytes from Mg batteries. L-AIE-F showed a strong “turn-on” AIE-active fluorescence at the leakage point of electrolyte from model Mg batteries. To the best of our knowledge, this is the first time that AIE technology has been used to sense the leakage of electrolytes.

## 1. Introduction

Magnesium (Mg) metal is an attractive anode material for rechargeable batteries, because it has a low reduction potential (−2.37 V vs. normal hydrogen electrode), a higher volumetric capacity than lithium and, unlike lithium, does not form dendrites during plating-stripping cycles [1,2,3,4,5]. Although lithium-ion batteries (LIBs) are widely used in portable electronic devices and electric vehicles due to the fact of their high energy density and long service life, the rapid consumption of LIBs is not sustainable due to the limited mineral resources of inorganic electrodes [6,7]. These issues limit the penetration of LIB technology into the large-scale energy storage market [8]. Since Mg is also inexpensive, highly abundant and environmentally benign, the Mg-metal rechargeable battery has long been viewed as a safe and low-cost alternative to the popular lithium-ion battery [9,10,11,12]. One of main drawbacks with Mg-metal batteries is the possible leakage of electrolytes, which are typically Grignard reagent, organoborate, borohydride, magnesium aluminate chloride complex, or Mg(TFSI)_2_-based solutions [13]. The leakage of these liquid electrolytes can cause many problems such as the corrosion of the metal casing of the battery [14], heavy metal ions in the electrolyte can cause environmental pollution [15], and fire and explosions can occur due to the leakage of flammable electrolytes [16]. Although there is thus an urgent need to develop a sensitive method for detecting leakage of electrolytes from Mg batteries, little attention has been paid to this problem. Fluorescence technologies are a good option for sensing, since they can provide fast, sensitive, and accurate analyses of guest species [17,18,19,20,21,22,23,24]. Fluorescent probes show a “turn-on” or “turn-off” fluorescence response to characteristic signal compounds in the analyzed guests [25,26,27,28,29,30,31,32,33,34,35]. The Mg^2+^ ion is one of the most abundant divalent ions, and they play a vital role in many chemical, biological, and environmental processes. In recent years, a variety of fluorescent probe have been developed by different research groups for Mg^2+^ detection. Suzuki et al. reported two new Mg^2+^ fluorescence imaging probes, KMG-20-AM and KMG-27-AM, both of which have a β-hydroxycarboxylate group and an aromatic amino group combined with a conjugated π-electron system, which will bring great changes in the fluorescence spectrum after forming a Mg^2+^ complex [36]. Ceroni and coworkers synthesized a hexathiobenzene molecule carrying six terpyridine units, and after adding of Mg^2+^ ions to the molecule in THF solution, metal-bridged crosslinking supramolecular polymer aggregates were formed, resulting in the observable turn-on phosphorescence [37]. Tang et al. report an efficient and convenient procedure for detecting Mg^2+^ with an AIE-active fluorescence probe in acetonitrile; this receptor showed a sensitive response to the addition of Mg^2+^ with enhanced fluorescence aggregation [38]. Among these probes, those based on aggregation-induced emission luminogen (AIEgens) are particularly attractive. AIE-active fluorescence was first reported by Tang’s group in 2001 [39]. Unlike traditional fluorescent chromophores, AIEgens become more emissive when aggregated [40,41,42,43,44]. Since AIEgen-based probes remain highly emissive in the aggregated or solid state, and they are readily portable and can conveniently be used as solids to sense guests, without the need for dissolution or other sophisticated pretreatments [45]. Motivated by these properties, here, we developed a novel naturally occurring AIEgen, laccaic acid (L-AIEgen), which can be extracted from the beetle Laccifer lacca. Compared with synthetic AIEgens, naturally occurring AIEgens are biocompatible, easily prepared and cheap [46,47,48]. The L-AIEgen showed a sensitive “turn-on” fluorescence to Mg^2+^. As a result, L-AIEgen was mixed with polyvinyl alcohol (PVA) to prepare composite films (L-AIE-F) for sensing leakage of electrolytes from Mg batteries (Figure 1). L-AIE-F showed a sensitive “turn-on” fluorescence when exposed to leakage of electrolytes (LX-144, 0.4 M (MgPhCl)_2_-AlCl_3_) from model Mg batteries and the detection limit was low at ~3.26 mmol.

## 2. Results and Discussion

The fluorescence of L-AIEgen in aqueous solution was very weak, but it intensified upon the addition of ethanol (Figure 2a and Table 1). When the fraction of ethanol reached 99%, the fluorescence intensity increased approximately six-fold, indicating AIE-active fluorescence of L-AIEgen. The absorption spectra of L-AIEgen were studied, and a red shift in the absorption peak was observed when ethanol was added, indicating the formation of J-aggregates (Appendix A) [48]. The addition of MgCl_2_ to an aqueous solution of L-AIEgen also enhanced the AIE fluorescence in a concentration-dependent manner (Figure 2b) and increased the fluorescence lifetime from 2.5 to 4.1 ns (Figure 2c). A wide variety of other cations were used to assess the selectivity of L-AIEgen, and none of these appreciably enhanced fluorescence (Appendix A). With the addition of Mg^2+^ ions into the L-AIEgen solution, coordination between Mg^2+^ and L-AIEgen occurred, resulting in fluorescence enhancement and a UV-vis absorption red shift (Appendix A). It might be attributed to the magnesium, which is preferred for forming a six-coordinated octahedral geometry by using N and O as ligands [49], the coordination of lone pairs of electrons on the N or O donor atoms to the Mg^2+^ sites, thereby stabilizing the excited state relative to the ground state, leading to longer wavelength absorption [50]. Encouraged by this high selectivity for Mg^2+^, we next evaluated the ability of L-AIEgen to sense LX-144, a typical Mg^2+^-containing electrolyte used in Mg batteries. The addition of LX-144 to a solution of L-AIEgen increased the fluorescence intensity in a concentration-dependent manner (Figure 2d). We compared the increase in the fluorescence intensity of L-AIEgen when the concentration of LX-144 and Mg^2+^ were the same, and we found that the fluorescence intensity of the LX-144 was not as good as Mg^2+^. This may be because the Mg in LX-144 exists in the form of (MgPhCl)_2_-AlCl_3_ complex, and its contact reaction with L-AIEgen was not as good as Mg^2+^, resulting in the fluorescence intensity of LX-144 being not as good as Mg^2+^ (Appendix A). In short, the spectra of L-AIEgen in the presence of increasing concentrations of LX-144 were similar to those in the presence of increasing concentrations of MgCl_2_, suggesting that fluorescence enhancement of L-AIEgen can be attributed to its reaction with Mg^2+^. All of these results demonstrate that L-AIEgen is, as expected, sensitive to LX-144, and that the sensitivity can be attributed to its reaction with Mg^2+^.

L-AIE-F was prepared by mixing L-AIEgen and PVA in aqueous solution, and its basic physical performance as a film was investigated. The SEM images (Appendix A) showed that laccaic acid was evenly distributed in the PVA matrix. Both laccaic acid and PVA molecules are rich in hydroxyl groups, which results in their high polarity and good compatibility. The appearance of L-AIE-F is shown in Appendix A. L-AIE-F can maintain a stable state in the ambient state, and it is still very stable after being placed in the air for 80 h. Its fluorescence spectrum is shown in Appendix A. Migration experiments using THF, and monitored with UV-Vis spectroscopy, showed that no L-AIEgen had leached into the THF, even after contact for 80 h (Figure 3a and Appendix A). Meanwhile, we also performed migration experiments using other solvents (i.e., water, ethanol, and ethyl ether) and monitored them using UV-Vis spectroscopy. After 80 h of exposure, no L-AIEgen was leached into these solvents (Appendix A), indicating that L-AIE-F was not only stable in THF, but also in water, ethanol, and ethyl ether. L-AIEgen was thus stably fixed in the PVA matrix, likely because of the hydrogen bonds between the hydroxyl groups of PVA and the phenolic groups of L-AIEgen. The mechanical performance of L-AIE-F was investigated next. The tensile strength and elongation at break were 44 MPa and 256%, respectively (Figure 3b and Appendix A). The tensile strength and elongation of PVA were 42 MPa and 269%, respectively (Figure 3b and Appendix A), showing that incorporation of L-AIEgen did not appreciably alter the mechanical strength of the PVA matrix, and the increase in the tensile strength of L-AIE-F (42 MPa to 44 MPa) indicated that there may be hydrogen bonds between the hydroxyl groups of PVA and the phenolic groups of L-AIEgen, which enhanced the interaction between PVA and L-AIEgen [51,52]. L-AIE-F was thus stable and had good mechanical performance. At the same time, we also measured the transmittance of L-AIE-F (Appendix A). After adding 0.1% wt laccaic acid to PVA, L-AIE-F still had a good transmittance, and the transmittance was still greater than 70% in the visible region (400–800 nm). The fluorescence of L-AIE-F was then measured in the presence of electrolyte containing Mg^2+^. L-AIE-F showed a concentration-dependent enhancement of fluorescence upon the addition of LX-144 (Figure 3c). Upon addition of Mg^2+^, the maximum fluorescence emission of L-AIE-F was at ~645 nm, representing a bathochromic shift compared with the fluorescence of L-AIEgen and LX-144 in solution (Figure 3d). The red shift in fluorescence might be attributable to the molecular J-type aggregation of L-AIEgen in the PVA matrix. Therefore, the UV absorption spectra of L-AIEgen and L-AIE-F were measured, and it was found that when L-AIEgen was in the PVA solute, the absorption peak showed an obvious red shift (Appendix A), from 488 to 523 nm, indicating the possible formation of J-aggregates [53]. Preliminary experiments were next carried out to investigate the sensitivity of the fluorescence emission of L-AIE-F to LX-144. The fluorescence emission of L-AIE-F showed marked enhancement upon the addition of LX-144 (Figure 3c,d). The relationship fitted the linear equation: y = 0.9x + 25 (R = 0.99), where the fluorescence is 645 nm measured at a given Mg^2+^ concentration (0–60 mm), and x is the concentration of Mg^2+^ added (Appendix A). The detection limit (3 s/K, s = standard deviation of the blank signal, K = 0.9) was ~3.26 mmol These results unambiguously confirmed that L-AIE-F was sensitive to LX-144, an electrolyte commonly used in Mg batteries.

Encouraged by the electrolyte-triggered enhancement of AIE, we next tested whether L-AIE-F could be used for fluorescence sensing of electrolyte leakage. To mimic Mg batteries, LX-144 electrolyte was placed in coin cell shells, with and without sealing rings, (Figure 4a,d), and the shells were then coated with L-AIE-F (Figure 4b,e). The fluorescence of L-AIE-F did not change noticeably when it was coated on the outside of coin cell shells with sealing rings (Figure 4c), but a strong enhancement in the fluorescence was observed when it was coated on the outside of shells without sealing rings (Figure 4f). These results unequivocally confirm that L-AIE-F could be used to sense leakage of electrolytes from Mg^2+^ batteries.

## 3. Materials and Methods

### 3.1. Materials

Laccaic acid was obtained from the Research Institute of Resources Insects, Chinese Academy of Forestry, Beijing, China. Poly (vinyl alcohol) (PVA, average degree of polymerization = 1750 ± 50) was purchased from Sigma-Aldrich, Shanghai, China. All other reagents and solvents were purchased from Merck Life Science Co., Ltd., Shanghai, China, or Shanghai Aladdin Bio-Chem Technology Co., Ltd., Shanghai, China. LX-144 electrolyte (0.4 M (MgPhCl)_2_-AlCl_3_ in THF) was purchased from Alibaba, Hangzhou, China.

### 3.2. Characterization

UV-Vis absorption spectra of L-AIEgen were recorded over the range 200–800 nm using a TU-1901 ultraviolet-visible double-beam spectrophotometer (Persee General Instrument Co., Ltd., Beijing, China). Photoluminescence (PL) was measured using a Fluo-max 4 spectrofluorometer (Horiba Scientific, Piscataway, NJ, USA). Tensile strength and elongation at break of L-AIE-F were measured using a UTM-2203 electromechanical universal testing machine (Suns Technology Stock Co., Ltd., Shenzhen, China). All measurements were performed at room temperature.

### 3.3. Preparation of L-AIEgen

Solution A: 10 mg laccaic acid was dissolved in 10.0 mL water to form 1 mg/mL laccaic acid solution.

### 3.4. Preparation of L-AIE-F

PVA (2.5 g) was dissolved in deionized water (50 mL), the solution was stirred magnetically for 2 h at 90 °C, and lac dye (2.5 mg) was then added. After stirring for a further 10 min, the mixture was poured onto a glass plate and dried naturally to give L-AIE-F. The films were dried at 30 °C and 50% humidity for 72 h before testing.

### 3.5. Sensing Electrolyte Leakage

LX-144 was placed in coin cell shells, with and without a sealing ring, to model intact and leaking Mg batteries. Then cut the L-AIE-F into a 2 × 2 cm square to completely coated onto these Mg battery models. After shaking the button battery, the electrolyte LX-144 in the Mg batteries without the sealing ring will leak. When the L-AIE-F film covering the outside of the Mg batteries contact with the leaked LX-144, the fluorescence of the film will increase accordingly. Electrolyte leakage was detected upon 365 nm UV irradiation.

## 4. Conclusions

In summary, we prepared a composite film (L-AIE-F) based on a naturally occurring AIEgen (L-AIEgen). In the presence of Mg^2+^, L-AIE-F showed a sensitive enhancement of AIE. This property allowed specific in situ detection of electrolyte leakage from a model Mg battery, demonstrating that L-AIE-F can be used practically for this purpose. Since L-AIE-F can be easily and cheaply prepared, it can be produced on a large scale and used commercially. In the future, L-AIE-F might be processed as a smart coating for Mg batteries, which can sense the leakage of electrolytes in situ. Additionally, following our strategy, AIEgen-based probes for other cations, such as Li^+^ and Fe^3+^ [54,55,56,57], might also be prepared as smart coatings to sense electrolyte leakage from Li or Fe batteries.

## Figures and Tables

**Figure 1 ijms-23-10440-f001:**
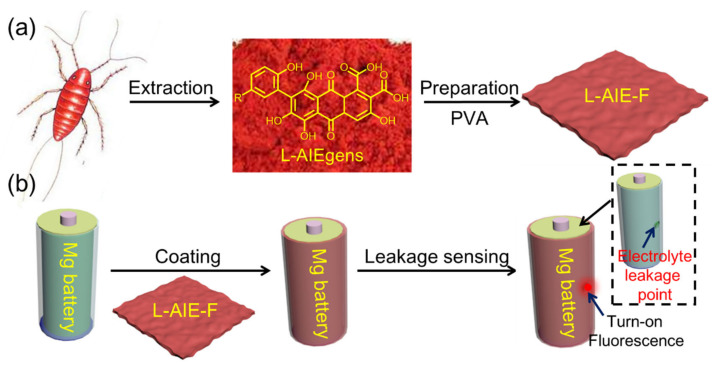
Schematic illustrations of the (**a**) preparation of L-AIE-F and (**b**) fluorescence sensing of electrolyte leakage from Mg batteries.

**Figure 2 ijms-23-10440-f002:**
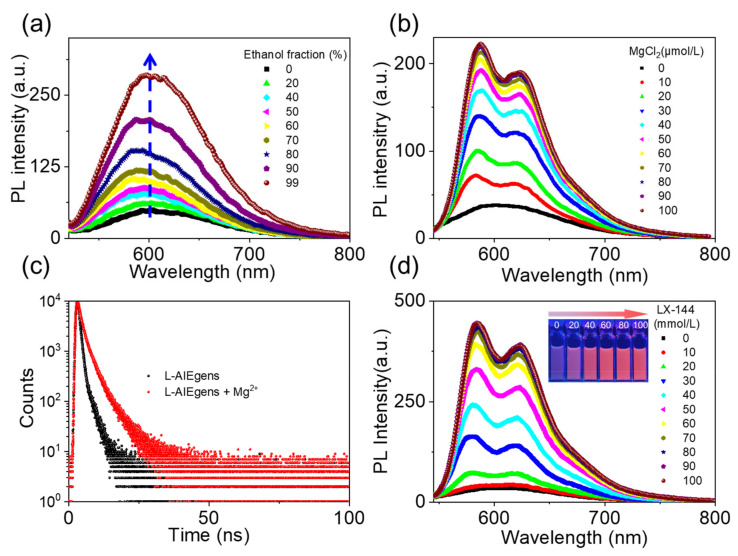
(**a**) Changes in the fluorescence of L-AIEgen in aqueous solution upon the addition of ethanol, with an excitation wavelength = 500 nm; (**b**) changes in the fluorescence of L-AIEgen (10 ppm) in ethanol solution upon the addition of MgCl_2_, with an excitation wavelength = 520 nm; (**c**) fluorescence lifetime of L-AIEgen in ethanol solution (10 ppm) in the presence and absence of Mg^2+^ (10 ppm), with an excitation wavelength = 520 nm; (**d**) changes in the fluorescence of L-AIEgen (10 ppm) in ethanol solution upon the addition of different volumes of LX-144 (0.4 m). PL Intensity (a.u.) = photoluminescence intensity (arbitrary units), with an excitation wavelength = 520 nm.

**Figure 3 ijms-23-10440-f003:**
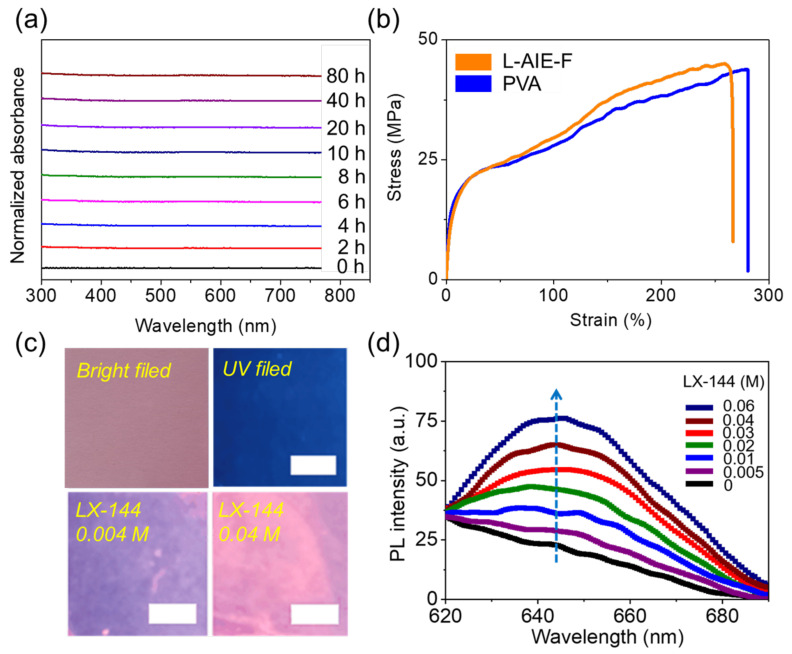
(**a**) In situ measurement of the absorbance of THF (10 mL) in the presence of L-AIE-F (2 × 2 cm) for different periods of time; (**b**) tensile strength of L-AIE-F and PVA; (**c**) images of L-AIE-F under bright field (**upper left**), UV field (**upper right**), UV field in the presence of 0.05 M LX-144 (**lower left**), and UV field in the presence of 0.4 M LX-144 (**lower right**); (**d**) fluorescence emission of L-AIE-F in the presence of different concentrations of LX-144, with an excitation wavelength = 365 nm.

**Figure 4 ijms-23-10440-f004:**
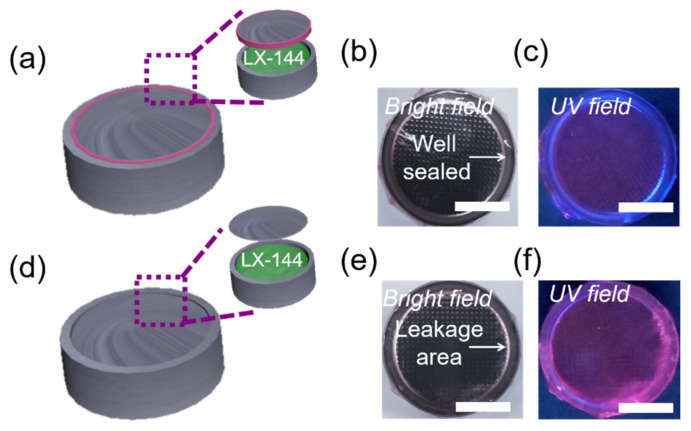
(**a**) Schematic illustration of well-sealed Mg battery model; images of the Mg battery model coated with L-AIE-F (**b**) in bright field and (**c**) upon UV irradiation (365 nm), scale bar = 0.5 cm; (**d**) schematic illustration of a leaking Mg battery model; images of a leaking Mg battery model coated with L-AIE-F (**e**) in bright field and (**f**) upon UV irradiation (365 nm), scale bar = 0.5 cm.

**Table 1 ijms-23-10440-t001:** Preparation of laccaic acid solutions with different ethanol fraction.

Ethanol Fraction (%)	Solution A (mL)	Water (mL)	Ethanol (mL)
0	0.1	9.9	0
20	0.1	7.9	2
40	0.1	5.9	4
50	0.1	4.9	5
60	0.1	3.9	6
70	0.1	2.9	7
80	0.1	1.9	8
90	0.1	0.9	9
99	0.1	0	9.9

## Data Availability

The data presented in this study are available upon request from the corresponding author.

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
