# Peer review of "Sensing Leakage of Electrolytes from Magnesium Batteries Enabled by Natural AIEgens"

_ijms, 2022, doi:10.3390/ijms231810440_

Round 1
Reviewer 1 Report
The authors adopt natural L-AIEgen to sense leakage of liquid electrolytes from magnesium (Mg) batteries. This manuscript is well organized. Such research is rare, and the date presented are interesting. Therefore, I suggest it be accepted after the following issues are addressed.
1. No L-AIEgen could leach into the THF. What about other solvents, such as ethanol, water and etc?
2. The selective sensing Mg2+ is quite interesting. What is the mechanism of sensing Mg2+ for L-AIEgen? The authors should discuss the sensing mechanism and provide sound evidence.
3. The authors state that “L-AIEgen was thus stably fixed in the PVA matrix, likely because of hydrogen bonds between the hydroxyl groups of PVA and the phenolic groups of L-AIEgen”. The hydrogen bondings should be characterized.
4. Fig S8 indicates that the response of L-AIEgen to Mg2+ is linear. The authors are encouraged to perform more titrations. The data presented here are not enough to prove the linear response.
5. More evidence should be provided to prove the J-type aggregation of L-AIEgen in the PVA matrix.
6. Errors: “ap-plication” in the abstract; excess blank in the “when it was coated on the outside of”; excess sentence “Electrolyte leakage was detected upon UV irradiation” in section 3.5 and etc.
Author Response
- No L-AIEgen could leach into the THF. What about other solvents, such as ethanol, water and etc?
A: Thanks for the significant question. We added leaching tests with other solvents (water, ethanol and ethyl ether) and also no L-AIEgen was leached. We added these results as Figure S9 in the revised manuscript.
- The selective sensing Mg2+ is quite interesting. What is the mechanism of sensing Mg2+ for L-AIEgen? The authors should discuss the sensing mechanism and provide sound evidence.
A: Thanks for the crucial suggestion. We added more discussion about the sensing mechanism in the revised manuscript. L-AIEgen contained -OH, -COOH and -NH groups, which possessed lone pair electrons. Upon addition of Mg2+ ions into the solution, L-AIEgen formed coordination with the O and N atoms, resulting in the absorption bathochromic and fluorescence enhanced. (Molecules 2020, 25, 3172; Dyes Pigm. 2004, 63, 141-150) We added the discussion in the revised manuscript and Figure S3 in the revised supporting information.
- The authors state that “L-AIEgen was thus stably fixed in the PVA matrix, likely because of hydrogen bonds between the hydroxyl groups of PVA and the phenolic groups of L-AIEgen”. The hydrogen bondings should be characterized.
A: Thanks for the significant suggestion. However, hydrogen bondings are very difficult to characterize, especially in this case that L-AIEgen was used in very little dosage. So we have to try other way to prove the hydrogen bondings. If the hydrogen bonding did not form between PVA chains and L-AIEgen, the tensile strength of the composite film should decrease compared to the pure PVA film. Here, we found that the tensile strength of pure PVA and L-AIE-F were 42 MPa and 44 MPa, respectively. It can be explained that the hydrogen bonds forming between the phenolic group of L-AIEgen and the hydroxyl group of PVA(ACS Appl. Mater. Interfaces 2021, 13, 5508-5517; Adv. Mater. 2020, 32, 1901244). We added the result as Figure S10 in the revised manuscript.
- Fig S8 indicates that the response of L-AIEgen to Mg2+ is linear. The authors are encouraged to perform more titrations. The data presented here are not enough to prove the linear response.
A: Thank you very much for the suggestion. We performed more titrations and added the new results as Figure 3d and Figure S13 in the revised manuscript.
- More evidence should be provided to prove the J-type aggregation of L-AIEgen in the PVA matrix.
A: Thanks for the significant suggestion. We measured the UV-vis absorption spectra of aqueous L-AIEgen and L-AIEgen in aqueous PVA. The results showed that the absorption of L-AIEgen in aqueous PVA had a significant red shift (488 nm to 523 nm), which proved that L-AIEgen forming J-type aggregation in aqueous PVA (J. Am. Chem. Soc. 2018, 140, 2727-2730). We added the results as Figure S12 in the revised manuscript.
- Errors: “ap-plication” in the abstract; excess blank in the “when it was coated on the outside of”; excess sentence “Electrolyte leakage was detected upon UV irradiation” in section 3.5 and etc.
A: Sorry about this. We corrected the mistakes.

Reviewer 2 Report
The manuscript describe Mg2+ detecting sensor by using the laccaic acid (LAIEgen). The aggregation-induced emission technology is helping to detect the leakage of liquid electrolytes in Mg- battery. The manuscript is interesting and it needs to address below comments.
1. Comparison of other materials for Mg2+ ion sensing will support this manuscript
2. It is suggesting to include in the introduction that “necessity of electrolyte ion (leakage) sensitivity in batteries ’’
3. In addition to THF, in a real Mg battery the common solvent is DME, so it would be useful to include absorbance test (leaching test) in the presence of L-AIE-F
4. Typo errors are noticed in multiple places some of them are given below. Please recheck the entire manuscript
a. In abstract : ap-plication
b. Affiliation : Universi-ty
c. Introduction : which are typically Grignard reagent-, organoborate-, borohydride-, magnesium aluminate chloride complex- or Mg(TFSI)2-based solutions
Author Response
1. Comparison of other materials for Mg2+ ion sensing will support this manuscript.
A: Thanks for the significant suggestion. We rewrote the introduction and added the comparison in the introduction part.
2. It is suggesting to include in the introduction that “necessity of electrolyte ion (leakage) sensitivity in batteries’’
A: Thanks for the suggestion. We rewrote the introduction and emphasized the necessity of electrolyte ion (leakage) sensitivity in batteries.
3. In addition to THF, in a real Mg battery the common solvent is DME, so it would be useful to include absorbance test (leaching test) in the presence of L-AIE-F.
A: I am sorry for that we could not obtain the solvent DME. So we have to use ethyl ether instead of DME. We added leaching tests with other solvents (water, ethanol and ethyl ether) and also no L-AIEgen was leached. We added these results as Figure S9 in the revised manuscript.
4. Typo errors are noticed in multiple places some of them are given below. Please recheck the entire manuscript
In abstract: ap-plication
Affiliation: Universi-ty
Introduction: which are typically Grignard reagent-, organoborate-, borohydride-, magnesium aluminate chloride complex- or Mg(TFSI)2-based solutions
A: We corrected the mistakes. Many thanks.

Reviewer 3 Report
In this paper, the authors reported the results of fabricating a film based on aggregation-induced emission luminogen (AIEgen) and testing it to detect an electrolyte leakage in Mg-metal batteries. Although the topic is thought to be novel and interesting, some corrections are required to be published in IJMS. My detailed comments are as follows:
1. (Title, Abstract) The dash in "Ena-bled" and "ap-plication" should be omitted.
2. (Abstract & Introduction) "AIEgen" is an abbreviation for "aggregation-induced emission luminogen". The abbreviation should be displayed after the full name. In addition, "the full name (abbreviation)" must be shown when it is first displayed in the text separately from the abstract.
3. (Introduction) "Electrolyte leakage" is described as a significant problem in Mg-metal batteries, and therefore the reason should be explained in more detail. Differences from other types of batteries (e.g. LiB) should also be described in the text.
4. Results and discussion on the durability of the L-AIE-F, such as stability to ambient air and moisture, are needed.
Author Response
- (Title, Abstract) The dash in "Ena-bled" and "ap-plication" should be omitted.
A: Sorry about this. We check the manuscript carefully and corrected all of them.
- (Abstract & Introduction) "AIEgen" is an abbreviation for "aggregation-induced emission luminogen". The abbreviation should be displayed after the full name. In addition, "the full name (abbreviation)" must be shown when it is first displayed in the text separately from the abstract.
A: Thanks for pointing this. We corrected the mistake in the revised manuscript.
- (Introduction) "Electrolyte leakage" is described as a significant problem in Mg-metal batteries, and therefore the reason should be explained in more detail. Differences from other types of batteries (e.g. LiB) should also be described in the text.
A: Thank you very much for the suggestion. We rewrote the introduction and emphasized the necessity of electrolyte ion (leakage) sensitivity in batteries. Also, we described the differences between magnesium-ion batteries and lithium batteries.
- Results and discussion on the durability of the L-AIE-F, such as stability to ambient air and moisture, are needed.
A: Thank you very much for the suggestion. We added the fluorescence changes of L-AIE-F at the ambient state in the Supporting Information as Figure S7 in the revised version.

Round 2
Reviewer 2 Report
The author has answered the quarries, and the manuscript can be accepted in its current form.